# The Combination of an mRNA Immunogen, a TLR7 Agonist and a PD1 Blocking Agent Enhances In-Vitro HIV T-Cell Immune Responses

**DOI:** 10.3390/vaccines11020286

**Published:** 2023-01-28

**Authors:** Lorena Usero, Lorna Leal, Carmen Elena Gómez, Laia Miralles, Elena Aurrecoechea, Ignasi Esteban, Berta Torres, Alexy Inciarte, Beatriz Perdiguero, Mariano Esteban, Felipe García, Montserrat Plana

**Affiliations:** 1AIDS Research Group, Institut d’Investigacions Biomèdiques August Pi i Sunyer (IDIBAPS), Hospital Clinic, University of Barcelona, 08036 Barcelona, Spain; 2Infectious Diseases Department, Hospital Clínic, University of Barcelona, 08036 Barcelona, Spain; 3Centro Nacional de Biotecnología (CNB), Department of Molecular and Cellular Biology, Consejo Superior de Investigaciones Científicas (CSIC), 28049 Madrid, Spain; 4CIBERINFEC, ISCIII—CIBER de Enfermedades Infecciosas, Instituto de Salud Carlos III, 28029 Madrid, Spain

**Keywords:** HIV, mRNA, vaccine, dendritic cells, immunomodulators, functional cure

## Abstract

The development of new strategies to achieve a functional cure for HIV remains a priority. We tested a novel HIV therapeutic vaccine using unmodified mRNA (TMEP-B) and mRNA modified by 1-methyl-3′-pseudouridylyl (TMEP-Bmod) expressing both a multiepitopic sequences from Gag, Pol, and Nef proteins, including different CD4 and CD8 T-cell epitopes functionally associated with HIV control in transfected monocyte-derived dendritic cells (MDDCs) obtained from HIV infected patients. In vitro assays were used to test the mRNAs alone and in combination with immunomodulator agents, such as the TLR-7 agonist Vesatolimod and the PD-1 antagonist Nivolumab to try to improve HIV-specific cellular immune responses. Combining the mRNAs with the immunomodulators enhanced HIV-specific T-cell responses, together with the secretion of IFNγ, IP10, MIP-1α, and MIP-1β, which are fundamental mediators of viral control. Our data suggest that the mRNA vaccine prototypes TMEP-B and TMEP-Bmod, when combined with Vesatolimod and/or Nivolumab, could achieve functional cure for patients with HIV.

## 1. Introduction

Antiretroviral therapy (ART) has significantly improved the morbidity and mortality associated with HIV [1]. However, it is unable to achieve cure or eradication. Thus, achieving functional cure remains the best approach, with promising results from several therapeutic vaccines. However, none has fully succeeded in controlling the viral replication of HIV.

Functional cure of HIV necessitates that a treatment induces not only an efficient immune response but also impacts the latent viral reservoir [2,3,4,5]. Nucleic acid (DNA/RNA) vaccines offer a novel and alternative strategy to conventional vaccines by directing the immune response against specific viral epitopes. Recently, our research group described the design and immunogenicity profile, in animal models, of a T-cell-based HIV-1 immunogen containing different domains of HIV-1 Gag, Pol, and Nef proteins expressed by DNA or modified vaccinia virus Ankara (MVA) vectors. This was named T-cell multiepitopic peptide (TMEP-B) [6,7]. Although DNA-based vaccines against HIV have been used in clinical trials, they have some clear limitations, such as the need for high doses when administered directly into the muscle, the need for electric devices to accelerate nucleic acid penetration into cells, and from having poor immunogenicity [8,9]. 

The direct in vitro administration of mRNA-based vaccine candidates into cells can lead to protein expression, providing new hope for a functional HIV cure [10,11]. These mRNA vaccines have many advantages with respect to efficient antigen expression efficiency. However, mRNA is fundamentally unstable, and the sequences and secondary structures that form (e.g., double helix, stem-loops structures) can be recognized by several innate immune receptors that could inhibit protein translation. Several methods with therapeutic relevance are being employed to improve the efficiency of translation, such as substituting modified nucleosides in the mRNA to make it non-inflammatory by avoiding type 1 IFN stimulation and allowing effective translation. It is also important for protein expression to achieve efficient intracellular delivery of mRNA in the cytosol, especially for mRNA administered systemically, to avoid its degradation and allow better cellular uptake and release. For this, lipid nanoparticles have been used successfully for mRNA release, even allowing for combinations of different mRNAs in a same formulation and improving the vaccine’s effectiveness [11,12,13,14,15,16,17,18]. 

The latent viral reservoir is a major obstacle to achieving functional cure of HIV. To overcome this drawback, researchers have proposed combining a therapeutic vaccine with latency reversal agents (LRAs) [19]. In this way, the combination of mRNAs encoding for HIV proteins with different immunomodulatory agents would result in an interesting strategy that could be developed to achieve functional cure. In non-human primates, using a TLR7 agonist in combination with other immunotherapies induced a state of remission and perhaps even the complete cure of HIV infection [20]. Moreover, using a TLR7 agonist together with HIV-1 peptides in vitro led to increased cell degranulation, cytokine production, and cytotoxic activity [21]. Agent inhibitors of the PD1/PDL1 pathway are other possible candidates, given that the expression of PD1 in infected cells contributes to the establishment and maintenance of HIV latency [22]. Furthermore, PD-1 blockade with a monoclonal antibody or the interference between PD1 and PDL1 during antigenic presentation can increase HIV-specific T-cells and enhance HIV latency reversal in patients who have achieve suppression with ART [22,23,24,25].

Using an in vitro model with cells from HIV infected patients, we evaluated the potential of unmodified or modified mRNA TMEP-B that coded for a multiepitope protein, either alone or in combination with a TLR7 agonist (Vesatolimod) or a PD1 antagonist (Nivolumab), to induce efficacious HIV-specific T-cell immune responses.

## 2. Materials and Methods

### 2.1. Study Individuals

This study included patients with chronic stable HIV infection and receiving ART with follow-up at the Clinic Hospital in Barcelona, Spain. Patients were eligible if they consented to donating blood and had a plasma viral load (pVL) of ≤50 copies/mL and a CD4+ T-cell count of >400/mm^3^. All patients signed an informed consent form agreeing to the use of their clinical data for research purposes. This study was reviewed and approved by the institutional Ethics review board (HCB/2018/0305).

### 2.2. Study Samples

Peripheral blood mononuclear cells (PBMCs) were isolated by Ficoll-Hypaque density-gradient centrifugation at 2000 rpm for 20 min at room temperature. The collected PBMCs were then washed twice with Dulbecco’s phosphate-buffered saline and manually counted for viability using a trypan blue exclusion dye. Freshly obtained PBMCs were used to generate monocyte-derived dendritic cells (MDDCs).

### 2.3. MDDC’s Generation

In order to obtain human monocytes, PBMCs were cultured in a 75 cm^2^ cell culture flask for 2 h at 37 °C in a 5% CO_2_ environment at a concentration of 3–4 × 10^6^ cells/mL in X-VIVO 15 media (Lonza, MD, USA) supplemented with 1% inactivated autologous serum, gentamicin (50 μg/mL, B/Braun Medical, Melsungen, Germany), fungizone (2.5 μg/mL, Life Technologies, California, USA), and Zidovudine (1μM, Genéricos Españoles Laboratorios, Madrid, Spain). Monocytes were confirmed to be in a monolayer after incubation for 2 h, and the nonadherent cells (MNC) were removed by three washes with phosphate-buffered saline (PBSx1) (Appendix A). Subsequently, monocytes were cultured for 6 days with 1000 U/mL each of recombinant human IL-4 (ProSpec, Rehovot, Israel) and recombinant human GM-CSF (ProSpec) on days 0 and 2. After 6 days, immature dendritic cells (iDCs) were obtained by adding cold PBSx1 to the 75 cm^2^ cell culture flask to the harvest them by firmly tapping the flasks. This dislodged the loosely adherent dendritic cell (DC) clusters. MDDCs were harvested into a 50 mL conical tube and counted manually for viability using the trypan blue exclusion dye. Lymphocytes were collected from MNC fractions and frozen as PBMCs for use in mixed lymphocyte response assays across different DC preparations.

### 2.4. Flow Cytometry

The phenotypes of MDDCs, iDCs, and mature DCs (mDCs), together with the expression of markers related to the migration ability of DCs and T-cell proliferation, were assessed by flow cytometry using different combinations of mAbs (Appendix A). The corresponding isotypes were used as controls. The viability and mortality of all cell populations were assessed using an Annexin V-PE/7-AAD Apoptosis Kit (Becton Dickinson) or the LIVE/DEAD™ Fixable Near-IR Dead Cell Stain Kit (ThermoFisher Scientific, Massachusetts, USA). Samples were acquired by FACSCanto II (BD Biosciences; San Jose, CA, USA) and analyzed with FlowJo (Tree Star, Ashland, OR, USA).

### 2.5. Transfection: Electroporation of MDDCs with Unmodified/Modified mRNA TMEP-B 

The TMEP-B protein sequence was translated into an RNA/codon-optimized nucleotide sequence avoiding all the time the RNA inhibition and instability elements, as well as intended splicing sites. The synthetic TMEP-B gene (1868 bp) was assembled from synthetic oligonucleotides and/or polymerase chain reaction products, and the fragment was inserted into the transfer vector pCyA-20 to generate the pCyA-20-TMEP-B plasmid (pc-DNA-TMEP-B). The production of unmodified (TMEP-B) and modified (TMEP-Bmod) mRNAs was carried out from the plasmid containing the TMEP multiepitope (pcDNA-TMEP-B) under the control of T7 polymerase (Trilink BioTechnologies, Inc.; San Diego, CA, USA) [6]. 

MDDCs were washed twice with IMDM medium, without serum, and centrifugated at 2000 rpm for 5 min to electroporate them with TMEP-B and TMEP-Bmod mRNA. Next, 4 × 10^6^ MDDCs for each electroporation condition were resuspended in 400 µL of the Ingenio^®^ Electroporation Kit and Solution (Mirus, Madison, WI), according to the manufacturer’s protocols. MDDCs electroporated without mRNA (i.e., mock transfected cells) were used as negative controls. The following electroporation settings were used for the mRNA: voltage 300 V, capacitance 150 Vf, and resistance 800 Ω (rack 0.4 µm). After electroporation, cells were transferred to fresh RPMI 1640 medium plus 10% fetal bovine serum and incubated for 2–3 h at 37 °C. After incubation, cells were resuspended in X-VIVO15 media (Lonza, MD, USA), supplemented with 1% inactivated autologous serum, gentamicin (50 μg/mL, B/Braun Medical, Melsungen, Germany), fungizone (2.5 μg/mL, amphotericin B, Life Technologies, California, USA), and Zidovudine (1 µM, Genéricos Españoles Laboratorios). Subsequently, electroporated DCs were seeded in a 96-well plate for 24 h in the presence of 1000 U/mL each of recombinant human IL4 (ProSpec), recombinant human GM-CSF (ProSpec), and 1000 U/mL of a maturation cocktail of cytokines. CellGenix provided tumor necrosis factor alpha (TNF-α), interleukin 6 (IL6) and interleukin 1 beta (IL1-β), while Pfizer supplied prostaglandin E2 (PGE2). After 24 h, electroporated DCs were collected for maturation analysis by flow cytometry and to carry out the co-culture.

### 2.6. Detection of Unmodified/Modified TMEP-B mRNA Expression by Flow Cytometry

TMEP-B and TMEP-Bmod expression in DCs was determined by flow cytometry 24 h after electroporation, using a specific antibody against the FLAG tag sequence located at the C-terminus of the TMEP-B sequence [6,7,26]. Cells were collected and washed with PBSx1 supplemented with 0.5% BSA and 0.1% sodium azide, before being centrifuged at 2000 rpm for 5 min. After surface staining (see above), cells were fixed/permeabilized with BD Cytofix/Cytoperm (BD Biosciences) at 4 °C for 20 min. Cells were then washed with PermWash buffer 1× (PW) (BD Biosciences) and centrifuged at 2000 rpm for 5 min. A blocking step was performed using an FcR Blocking Reagent (Milteny Biotec, Bergisch Gladbach, Alemania) for 30 min at 4 °C. After blocking, cells were washed twice with PW 1× and centrifuged at 2000 rpm for 5 min. Next, resuspended cells were incubated with 5 µg/mL of the monoclonal antibody anti-FLAG M2 (Sigma-Aldrich, Burlington, MA, USA) in PW 1× at 4 °C in the dark for 45 min. The cells were then washed twice with PW 1× and the secondary anti-mouse IgG (H + L) Alexa488 antibody diluted (1:400, Invitrogen, Carlsbad, CA, USA) in PW 1× was added to the cells at 4 °C for 45 min in the dark. After incubation, cells were washed twice with PW 1× and fixed with 1% formaldehyde. Samples were acquired with a FACSCanto II flow cytometer (BD) and data analyses was completed using FlowJo (Version 10.4.2; Tree Star, Ashland, OR, USA).

### 2.7. Blocking PD1 Expression on T-Cells

PD1 expression on T-cells was blocked by incubating with 20 µg/mL Nivolumab (anti-PD1, MedChemExpress, NJ, USA) for 30 min at 37 °C. After incubation, cells were washed twice with PBSx1 supplemented with 0.5% BSA and 0.1% sodium azide, then washed once with PBSx1 before being centrifuged at 1500 rpm for 5 min. Next, cells were incubated with Fc Block (Milteny Biotech, Bergisch Gladbach, Alemania) for 30 min at 4 °C and then washed with PBSx1 supplemented with 0.5% BSA and 0.1% sodium azide before being centrifuged at 1500 rpm for 5 min. The cells were then subjected to surface staining using anti-nivolumab PE (Anti-Human IgG4 pFc, Abcam, Cambridge, United Kingdom), anti-PD1 PE-Cy7 (Clone: EH12.1, BD Pharmingen, San Jose, CA, USA), and CD3 PERCP (Clone: HIT3a, BD Pharmingen) [27]. The corresponding isotypes and tube with cells without previous nivolumab incubation were used as controls. PD1 expression and nivolumab binding through human IgG4 were evaluated by flow cytometry using FACSCanto II (BD Biosciences) and analyzed using FlowJo (Tree Star, Ashland, OR, USA).

### 2.8. Co-Cultures: Electroporated DCs with Autologous Lymphocytes

The stimulatory function of the electroporated DCs was assessed by their ability to induce in vitro proliferation of autologous nonadherent PBMCs used as a source of enriched T-cells. These T-cells (2 × 10^5^ cells/well) were incubated in X-VIVO 10 medium with the DCs (6 × 10^4^ cells/well) previously electroporated with TMEP-B or TMEP-Bmod mRNA, and then they were seeded in 96 round-well plates at 37 °C in a humid 5% CO_2_ atmosphere at a ratio of 1 DC to 3 T-cells. Mock electroporated DCs were used as negative controls. To determine the roles of Nivolumab (Anti-PD1) and Vesatolimod (TLR7 agonist) on T-cell proliferation, we added 20 µg/mL of Nivolumab (MedChemExpress) and 1000 nM of Vesatolimod into the DC/T-cells co-culture either separately or together. Both agents were added when setting up the co-cultures that also contained the cytokine cocktails. To assess T-cell proliferation, these were stained with carboxyfluorescein succinimidyl ester (CFSE) using the Cell Trace CFSE proliferation kit (Invitrogen) according to the manufacturer’s instructions. T-cell proliferation was measured by flow cytometry of CFSE dilution after 6 days of co-culture and expressed as the percentage of CFSE^low^ cells after 6 days of co-culture under the different conditions.

### 2.9. Cytokine Secretion Analysis

The co-culture supernatants were collected after both 48 h and 6 days, and then stored at −20 °C for further cytokine and chemokine quantification. Cytokine/chemokine secretion was measured using Luminex (Cytokine Human 25-Plex Panel; Invitrogen), according to the manufacturer’s protocol. The following 25 mediators were tested: Eotaxin, GM-CSF, IL-1β, IL-1RA, IL-2, IL-2R, IL-4, IL-5, IL-6, IL-7, IL-8, IL-10, IL-12p40/p70, IL-13, IL-15, IL-17, IFN-α, IFN-γ, IP-10, MCP-1, MIG, MIP-1α, MIP-1β, RANTES, and TNF-α.

### 2.10. Statistical Analysis

Cytometry results are shown as the means ± standard deviation (SD) of percentages or as the mean fluorescence intensity (MFI). Data analysis and comparison of each parameter relied on Student *t*-tests and nonparametric Wilcoxon signed rank tests, as appropriate. Statistical analysis was performed using GraphPad Prism, version 5 (La Jolla, CA, USA). For all analyses, the level of significance was set at * *p* < 0.05, ** *p* < 0.01 and *** *p* < 0.001.

## 3. Results

### 3.1. TMEP-B Affected DCs Maturation

We analyzed the phenotype of the DCs and found a high expression of CD86 (≥80%; MFI, 340), CD40 (98%; MFI, 501), HLADR (98%; MFI, 1172), and PDL1 (65%; MFI, 387). By contrast, the levels of CD80 (MFI: 261) and CD83 (MFI: 165) were both below 10% (Appendix A). When analyzing the effect of TMEP-B and TMEP-Bmod on DC maturation after electroporation (Figure 1A,B), we found high CD86 expression with both mRNAs (TMEP-B, 86.83% ± 18.83%; TMEP-Bmod, 88.61% ± 21.73%) and in the mock transfected DCs (85.08% ± 19.64%). However, the expressions of CD80 (TMEP-B, 29.04% ± 22.38%; TMEP-Bmod, 23.59% ± 22.68%; mock, 25.74% ± 19.01%) and CD83 (TMEP-B, 48.58% ± 23.83%; TMEP-Bmod, 49.25% ± 22.91%; mock, 46.36% ± 20.80%) increased after DC electroporation, but this was lower than the CD86 expression and marginally higher than in the mock transfected cells (Figure 1B). Cell viability was 95–100% in all cases (Data not shown).

### 3.2. Flag Expression on DCs Transfected with TMEP-B and TMEP-Bmod 

Flag expression was similar for DCs electroporated with any mRNA (TMEP-B: 39.07% ± 25.84; MFI: 446 and TMEP-Bmod: 31.86% ± 22.61; MFI: 431) whereas no Flag expression was detected in mock DCs (2.52% ± 4.04; MFI: 233) (Figure 2A,B). When comparing Flag expression on DCs electroporated with mRNAs and mock transfected DCs, we found statistically significant differences (mock vs. TMEP-B, *p* = 0.0007 and mock vs. TMEP-Bmod, *p* = 0.0008) (Figure 2B).

### 3.3. CCR7 Expression on iDCs and mDCs after mRNA Transfection

Our results showed a low CCR7 expression level on iDCs (15.74% ± 10.46%) (Appendix A). When repeated on mDCs (Appendix A), CCR7 expression was higher than on iDCs. The DCs that had been transfected with any mRNA showed higher levels of CCR7 expression (TMEP-B: 39.1% ± 29.79%, TMEP-Bmod: 35.29% ± 31.44%) compared to mock transfected cells (28.23% ± 11.48%) but differences were not significant.

### 3.4. T-Cells Proliferation Induced by mRNA Electroporation

T-cell proliferation was slightly higher when T-cells were co-cultured with DCs that had been electroporated with mRNA than in the mock scenario for both CD4+ T-cells (TMEP-B, 2.73 ± 2.23; TMEP-Bmod, 3.65 ± 4.93; mock, 2.01 ± 1.27) and CD8+ T-cells (TMEP-B, 2.17 ± 1.66; TMEP-Bmod, 3.03 ± 3.08; mock, 1.93 ± 1.58) (Figure 3A). Differences were statistically significant for CD8+ T-cells proliferation with TMEP-B (*p* < 0.05) and TMEP-Bmod (*p* < 0.01). Moreover, 48 h after co-culture, we analyzed the IL2 secretion in the culture supernatant and found that it was significantly higher between conditions where the DCs were electroporated with TMEP-B versus mock (*p* < 0.05) and with TMEP-B versus TMEP-Bmod (*p* < 0.05) (Figure 3B).

### 3.5. Cytokine and Chemokine Secretion Induced by mRNA 

IFNγ secretion on day 6 of co-culture was significantly higher for DCs electroporated with TMEP-B (*p* = 0.0020) and TMEP-Bmod (*p* = 0.0391) than for those in the mock scenario (Figure 4A). We also analyzed cytokines and chemokines related to DC function and the control of viral infection, finding differences in IP10, IL12, CCL5, MIP-1α, and MIP-1β (Figure 4B–F). The secretions of IL12 (*p* = 0.0391), MIP-1α (*p* = 0.0273), and MIP-1β (*p* = 0.0391) were also increased when using DCs electroporated with TMEP-B. By contrast, neither IP-10 (CXCL10) and CCL5 (Rantes) (Figure 4B,D) nor IFNα, IL6, and TNFα (Appendix A).

### 3.6. Combining TMEP-B with Nivolumab and Vesatolimod Improved HIV-Specific T-Cells Proliferation

We started by assessing the effects of both nivolumab and vesatolimod in cellular assays, confirming that nivolumab caused a loss of PD1 expression on T-cells (Figure 5A) and vesatolimod increased IFNα, IL6, TNFα, and IL1β secretion in the TLR-7 pathway (Appendix A). After evaluating the effects of these immunomodulators, we analyzed the proliferation of HIV-specific CD4+ and CD8+ T-cells after co-culture with TMEP-B transfected DCs after adding vesatolimod, nivolumab, or both. CD3+ CD4+ T-cell proliferation was similar in all conditions (TMEP-B, 1.97% ± 0.98%; TMEP-Bmod, 1.75% ± 1.02%; mock, 2.43% ± 1.42%). However, CD4+ T-cell proliferation increased in all cases after adding vesatolimod (TMEP-B, 3.08% ± 2.13%; TMEP-Bmod, 3.12% ± 2.36%; mock, 2.91% ± 1.91%), although the difference was greater when using mRNA-electroporated DCs (Figure 5B).

Furthermore, we observed a significant increase in proliferation when combining TMEP-B with nivolumab (4.28% ± 3.41%) alone or with both nivolumab and vesatolimod together (3.81% ± 2.67%) (Figure 5B). We compared the effect on T-cell proliferation with TMEP-B alone or in combination with one or both immunomodulators. This revealed a significantly higher proliferation when combined with nivolumab alone (*p* = 0.0313) or with both nivolumab and vesatolimod (*p* = 0.0156). We also compared different combinations of TMEP-B with the following: vesatolimod versus nivolumab (*p* = 0.0124); vesatolimod versus nivolumab and vesatolimod combined (*p* = 0.0039). This effect was not seen using TMEP-Bmod. Comparing different strategies, we found that the combination of TMEP-B and nivolumab induced better CD4+ T-cell proliferation than combining this immunomodulator with TMEP-Bmod (*p* = 0.0156) or with the mock (*p* = 0.0039) (Figure 5B, left).

CD3+ CD8+ T-cell proliferation levels at baseline were as follows: TMEP-B, 2.11% ± 1.23%; TMEP-Bmod, 2.08% ± 1.27%; and mock, 1.82% ± 0.84%. Combining TMEP-B with vesatolimod or nivolumab caused the proliferation level to increase by 0.34% or 1.22%, respectively. When both agents were combined with TMEP-B, the proliferation level was 2.87% ± 2.45%, lower than when combined with only nivolumab (3.33% ± 2.78) but higher than when combined with only vesatolimod (2.45% ± 2.18; Figure 5B). We found significant differences when comparing the following: TMEP-B against TMEP-B with nivolumab (*p* = 0.0104), TMEP-B with nivolumab versus TMEP-B with vesatolimod (*p* = 0.0071); and TMEP-B with vesatolimod versus TMEP-B with vesatolimod and nivolumab (*p* = 0.0398). We did not see these differences with TMEP-Bmod. Differences existed when comparing TMEP-B against mock, with both exposed to nivolumab (*p* = 0.0071) (Figure 5B, right).

### 3.7. Cytokines and Chemokines That Promote Viral Control Increase after TMEP-B or TMEP-Bmod Are Combined with Nivolumab or Vesatolimod

The secretions of IFNγ and MIP-1β increased, as did IP-10 to a lesser degree, after combining TMEP-B or TMEP-Bmod with vesatolimod (Figure 5C). This effect on IFNγ secretion was also observed for DCs transfected with TMEP-B or TMEP-Bmod when combined to nivolumab alone or with both nivolumab and vesatolimod (Figure 5C). The use of nivolumab in combination with different agents had a significant impact on IFNγ secretion (Figure 5C). This effect of nivolumab was also seen for MIP-β secretion (Figure 5C). In the case of IP10 secretion, this effect was only seen when combining nivolumab with TMEP-B (*p* = 0.0078) and compared with TMEP-B alone or TMEP-B plus vesatolimod (*p* = 0.0156) (Figure 5C).

## 4. Discussion

The development of an effective therapeutic vaccine against HIV depends not only on understanding the complexity of the immune responses against the virus but also its adaptation to selective pressure exerted by the host. In this study, we evaluated an immunogen against HIV-1 that encodes a multiepitopic T-cell protein, which has been named TMEP-B. In previous studies, vectoring TMEP-B in both DNA and MVA enhanced HIV-1 specific T-cell responses, supporting its potential utility in prime/boost strategies for HIV therapy [6]. In the present study, we have evaluated the effect of this TMEP-B immunogen on HIV-specific T-cells immune responses, using unmodified and modified mRNA vectors.

TMEP-B and TMEP-Bmod similarly increased CD86, CD80, CD83, and FLAG expression levels, suggesting that the chemical modification made in the mRNA did not affect DC maturation or mRNA translation. Similar results have been obtained in a recently published study using HEK293T cell line [26]. In this, FLAG expression determined by flow cytometry or Western blot at different time-points was higher when transfected with TMEP-B, suggesting that differences in FLAG expression depend on the technique, experimental approach, and cell type used, as well as the time in which protein expression is determined. This further corroborates our finding that chemical modification of mRNA did not seem to improve translation. In addition, CCR7 expression was increased in DCs electroporated with any of our mRNAs, showing the potential impact of each immunogen on DC migration to lymph nodes.

Analysis of how mRNAs affected T-cell proliferation revealed that both TMEP-B and TMEP-Bmod can increase proliferation. This was also related to an extremely mild in vitro increase in IL2 secretion. We found no differences in T-cell proliferation after transfecting DCs with TMEP-B or TMEP-Bmod, which could be explained by the method used to transfect cells. When electroporation is used for transfection, most of the material goes straight into the cytosol, thereby avoiding degradation in the endocytic pathway [13].

It should also be noted that mRNA vaccines delivered by lipid nanoparticles would be more relevant than electroporation for both in vitro and in vivo human experimentation. Therefore, we are now formulating our mRNA TMEP-B in lipid nanoparticles to increase their efficiency, expression, and stability for further testing.

TMEP-B and TMEP-Bmod both stimulated cytokine and chemokine secretion. Indeed, the IFNγ secretion important for viral control was similar using either TMEP-B or TMEP-Bmod, suggesting that this was probably independent of nucleoside modification in the mRNA. A recent publication of the impact of TMEP-B and TMEP-Bmod on cytokine and chemokine secretion notably found an increase in IFIT1 and IFIT2 secretion (Interferon-induced protein) at 6 h post-transfection when using RT-qPCR. Such data would indirectly corroborate the observed increase in IFNγ levels. However, they found differences when using TMEP-B and TMEP-Bmod for IFIT1 and IFIT2 secretion, with the TMEP-B triggering higher levels of IFIT1 than TMEP-Bmod in the THP-1 cell line and enhancing the expression of IFIT2 [26]. In our head-to-head comparison of both mRNAs, we found higher levels of certain cytokines and chemokines when using TMEP-B (e.g., IP10, MIP-1α, and MIP-1β). These have been described as major HIV suppressive factors produced by CD8+ T-cells [28]. Our data therefore suggest that other elements of mRNA design, not the chemical modification, could be involved in the observed immunogenic effects. Other studies performed in non-human primates have also showed that the optimization of non-coding mRNA elements could improve the immunogenicity and protective efficacy of mRNA vaccines [29,30]. This could be the case for TMEP-B mRNA in this research.

Adjuvants or immunomodulators given systemically could be used in combination with TMEP-B to improve antigen presentation and the specific host immune response. We used vesatolimod, a promising candidate that could combine with other immunotherapies to achieve a functional cure of HIV. Our results show that vesatolimod use in combination with the mRNAs led to increased T-cell proliferation for both CD4+ and CD8+ T-cells. Vesatolimod also favored IFNγ secretion and the production of other cytokines and chemokines (e.g., IP-10 or MIP-1β). It could be speculated that various cellular antiviral responses are induced upon the activation of TLR7, and that this includes the release of inflammatory cytokines. Furthermore, it is possible that TLR7 activation by vesatolimod could promote cross-presentation via antigen-presenting cells to enhance the CD8+ T-cell responses. Vesatolimod is described as a modulator of HIV-1 that activates immune cells in vitro. It has been described that combining a TLR-7 agonist with HIV peptides (derived from Gag, Pol, Env, and Nef) can increase CD8+ T-cell degranulation, cytokine production, and cytolytic activity [21]. Studies in non-human primates have demonstrated the ability of TLR7 agonists to induce potent immune responses with increased IFN-regulated antiviral gene expression [31]. Elsewhere, it has been shown that the combination of Ad26/MVA vaccination and TLR7 stimulation can improve virologic control and delayed viral rebound after interruptions to antiretroviral treatment [20]. Recent clinical trials of Vesatolimod in HIV infected individuals have shown that this produced a consistent cytokine response, interferon-stimulated gene expression, and lymphocyte activation [32,33].

The other immunomodulatory agent evaluated in this study was nivolumab, an inhibitor of the PD1/PDL1 pathway. Different studies have associated PD1 expression on the T-cells of HIV infected patients with eventual T-cell exhaustion and disease progression [34,35]. CD4+ T-cells expressing immune checkpoint inhibitors have also been found to be enriched in latent HIV-1, suggesting that PD1 is important in establishing HIV latency [22,25]. Thus, we envisage that using nivolumab during antigen presentation at the immune synapse could be used to rescue T-cells from exhaustion and anergy, while also avoiding the undesirable systemic effects of blockade and possibly boosting anti-virus immunity. In fact, the expression, secretion, and binding of these soluble molecules (PD1/PDL1) after mRNA electroporation, favoring T-cell multifunctionality, has been previously demonstrated [23]. Our results showed that combining nivolumab with TMEP-B increased not only the proliferation of HIV-specific T-cells (both CD4+ and CD8+) and cytokines (IFNγ and IP10) but also that of other cytokines and chemokines with antiviral effects (e.g., MIP-1β). These data suggest that nivolumab improves the specific immune response against HIV infection. Studies in non-human primates with the simian immunodeficiency virus have shown that PD1 blockade with an antibody led to a rapid expansion of virus-specific CD8+ T-cells with improved functional quality, improving B-cell proliferation and in increasing envelope-specific antibodies to the virus, thereby enhancing both cellular and humoral immune responses [36]. Other studies in non-human primates have shown improved antiviral function of CD8+ T-cells and B cells, together with a reduction in the size of the viral reservoir that led to improved control of viral rebound after ART interruption [36,37,38].

Finally, combining both immunomodulatory agents, vesatolimod and nivolumab, in our experiments did not increase the levels of CD4+ and CD8+ T-cell proliferation compared with the use of either agent alone. Research must now investigate the effects of both agents to determine the impact of our strategy on T-cell polyfunctionality.

Overall, we have shown that combining mRNA-based vaccines with immunomodulatory agents can enhance immunogenicity and increase the specific cellular immune response against HIV infection. Our findings suggest that the proposed combination approach represents a promising strategy that could deliver functional cure of HIV. Moreover, in the future our experimental approach could be used as a helpful platform to screen more and different immunogenic combinations. Further research is clearly warranted to tackle these possibilities.

## Figures and Tables

**Figure 1 vaccines-11-00286-f001:**
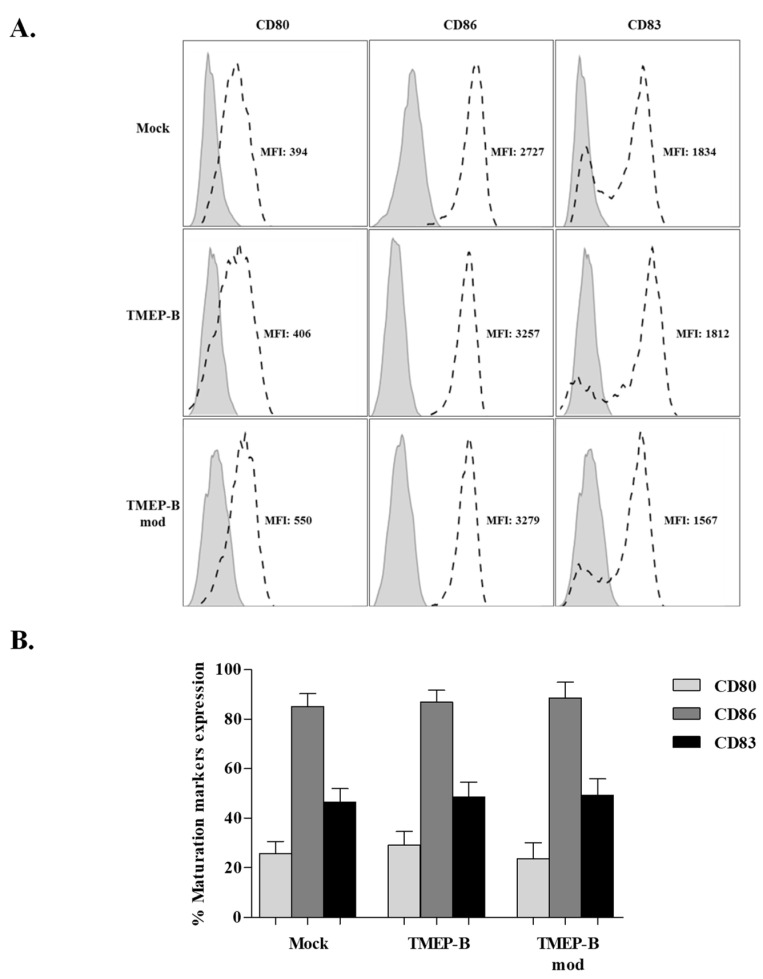
Analysis of maturation markers on DCs from HIV infected patients electroporated with mRNA TMEP-B or TMEP-Bmod. Analysis of maturation markers levels (CD80, CD86, and CD83. (**A**) Histogram showing the mean fluorescent intensity (MFI) in a representative example of all experiments (*n* = 17). (**B**) Percentages of CD80, CD86 and CD83 expression on DCs from HIV infected patients electroporated with TMEP-B and TMEP-Bmod (*n* = 17). Mock transfected DCs were used as negative controls and pairwise comparisons were made (Mock vs. TMEP-B; Mock vs. TMEP-Bmod; TMEP-B vs. TMEP-Bmod). A cytokine maturation cocktail was added to all co-cultures. Percentages ± SD are shown. Analysis by the Wilcoxon signed rank test (* *p* < 0.05; ** *p* < 0.01; *** *p* < 0.001).

**Figure 2 vaccines-11-00286-f002:**
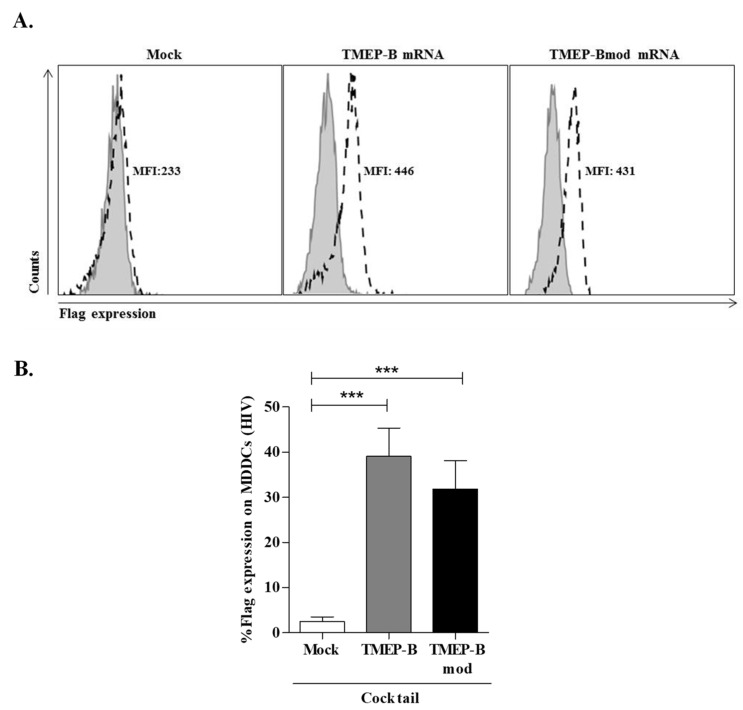
FLAG expression analysis on DCs from HIV infected patients transfected with TMEP-B or TMEP-Bmod mRNA by flow cytometry. DCs were electroporated with 10 µg of TMEP-B or TMEP-Bmod. Mock transfected cells were used as a negative control. At 24 h post-transfection, cells were collected and processed for flow cytometry analysis. FLAG expression was analyzed using 5 µg/mL of monoclonal antibody anti-FLAG M2. (**A**) Histogram showing the mean fluorescent intensity (MFI) in a representative experiment. FLAG expression was determined on Mock transfected cells and DCs transfected with TMEP-B and/or TMEP-Bmod. (**B**) Percentage of FLAG expression on DCs (from HIV infected patients) electroporated with TMEP-B and TMEP-Bmod, and on mock transfected cells (*n* = 17). The cytokine cocktail maturation was added to all cultures. Analysis was by the Wilcoxon signed rank test (**p* < 0.05; ***p* < 0.01; ****p* < 0.001).

**Figure 3 vaccines-11-00286-f003:**
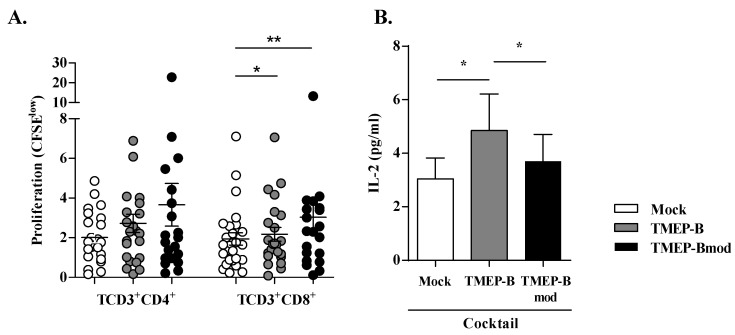
Analysis of HIV-specific T-cell proliferation in co-cultures using DCs electroporated with mRNA TMEP-B or TMEP-Bmod. T-cell proliferation after 6 days of co-culture (DC:T-cells) using DCs from HIV infected patients and electroporated with TMEP-B or TMEP-Bmod. (**A**) Evaluation of HIV-specific T-cell proliferation (CD3+ CD4+ and CD3+ CD8+) using DCs transfected with 10 µg of TMEP-B or TMEP-Bmod (*n* = 26). (**B**) Analysis of IL2 secretion in the supernatant 48 h after co-culture (DC:T-cells) using DCs electroporated with TMEP-B or TMEP-Bmod (*n* = 8). The cytokine maturation cocktail was added to all co-cultures. Percentages ± SD are represented. Analysis was by the Wilcoxon signed rank test (**p* < 0.05; ***p* < 0.01; ****p* < 0.001).

**Figure 4 vaccines-11-00286-f004:**
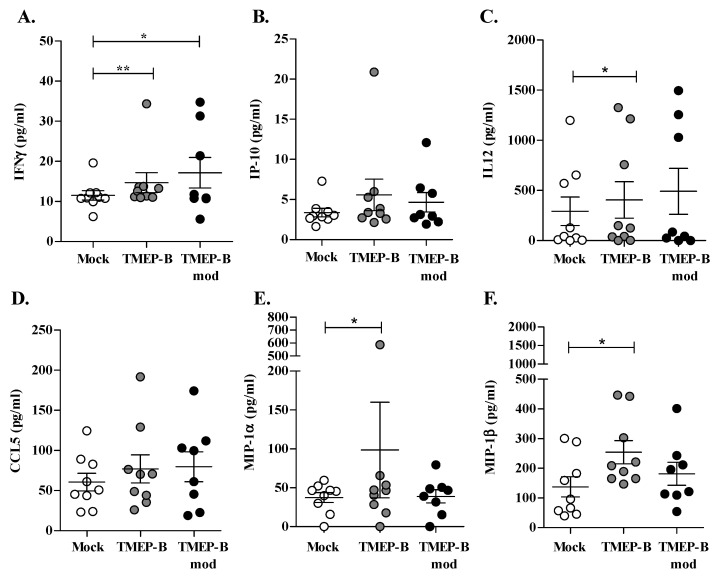
Cytokine secretion in the co-cultures of DC-T-cells in HIV infected patients. (**A**–**F**). Analysis of IFNγ, IP-10, IL12, CCL5, MIP-1α, and MIP-1β secretion after 6-days of co-cultures (DCs: T-cells), using DCs from HIV infected patients and electroporated with TMEP-B (gray circle) or TMEP-Bmod (black circle) (*n* = 9). Mock transfected DCs were used as negative controls (white circle). Percentages ± SD are represented. Analysis was with the Wilcoxon signed rank test (* *p* < 0.05; ** *p* < 0.01; *** *p* < 0.001).

**Figure 5 vaccines-11-00286-f005:**
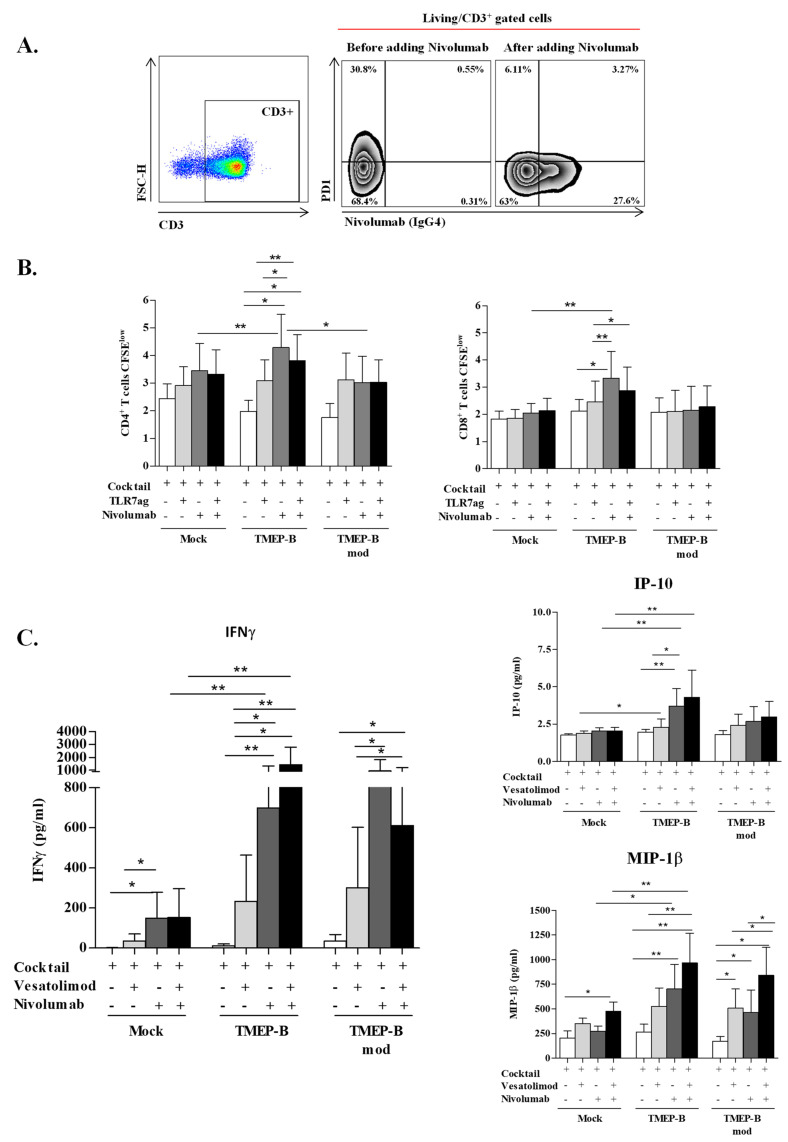
HIV-specific T-cell proliferation and cytokine secretion with or without vesatolimod or nivolumab. (**A**) Analysis of PD1 expression by flow cytometry using T-cells from HIV infected patients and a specific anti-PD1 antibody (EH12.1) with or without Nivolumab (20 µg/mL). (**B**) Analysis of T-cell proliferation (CD3+ CD4+ and CD3+ CD8+) in co-cultures using DCs electroporated with TMEP-B or TMEP-Bmod and adding only cocktail cytokines or in combination with nivolumab (20 µg/mL), vesatolimod (1000 nM) or both (*n* = 8). (**C**) Analysis of cytokine secretion (IFNγ, IP10, MIP-1β) after 6 days of co-culture using DCs electroporated with TMEP-B or TMEP-Bmod (*n* = 9). Mock transfected DCs were used as negative controls. Graphic shows the cytokine secretion in the following settings: cocktail added in the co-culture (white); cocktail plus 1000 nM vesatolimod (light gray); cocktail plus 20 µg/mL nivolumab (dark gray); or cocktail plus both agents (Black). Percentages ± SD are shown. Analysis was by Wilcoxon signed rank test (* *p* < 0.05; ** *p* < 0.01; *** *p* < 0.001).

## Data Availability

The datasets generated during and/or analyzed during the current study are available from the corresponding author on reasonable request.

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
