# Peer review of "The Combination of an mRNA Immunogen, a TLR7 Agonist and a PD1 Blocking Agent Enhances In-Vitro HIV T-Cell Immune Responses"

_vaccines, 2023, doi:10.3390/vaccines11020286_

Round 1
Reviewer 1 Report
In this study the authors aimed to evaluate the potential of unmodified or modified mRNA TMEP-B coding for a multiepitope protein, alone or in a combination with a TLR7 agonist (Vesatolimod® ) or an anti-PD1 (Nivolumab® ) to induce efficacious HIV specific T cell immune responses in an in-vitromodel using cells from HIV-infected patients. They
have shown the relevance of combining mRNA-based vaccines with immunomodulatory agents to enhance immunogenicity and increase the specific cellular immune response against HIV infection. The obtained findings suggest that these combination proposals could be a promising strategy to test to achieve a HIV functional cure. In addition, in the future our experimental approach could be used as a helpful platform to screen more and different immunogenic combinations.
The manuscript is well prepared and reads good. The quality of results presentation is high. The idea to perform such a study seems interesting and the obtained results are promising to find a curative vaccination against HIV infection.
I have no significant queries and I do recommend the acceptance of the manuscript in the present form.
Reviewer 2 Report
Comments on the manuscript “The combination of an mRNA immunogen, a TLR7 agonist and a PD1 blocking agent enhances in-vitro HIV T-cell immune responses” by Usero L et al.
In this study, authors used PBMC cells isolated from HIV-infected patients as a primary source to test ex vivo effect of two types of mRNA vaccine agents on proliferation and stimulation of CD4+ and CD8+ in combination with several types of immunomodulators.
I address authors several following comments:
11. In Figure 5A the authors demonstrate effect of Nivolumab binding to CD3+ gated T-cells shown as the decrease of anti-PD-1 monoclonal antibody EH12.1 binding. However, to interpret such effect, authors should provide evidence that both monoclonal antibodies compete for an overlapping binding site on the extracellular domain of PD-1. While the binding site for Nivolumab is known from the structure of PD1/PDL1 complex (PDB ID 5ggr), demonstration of the binding site for EH12.1. is not documented by the authors. If this information is known, authors should provide a relevant reference. If the binding site is not known, the authors should provide an evidence that these both antibodies compete for the binding site. This could be done by SPR, MST, LigandTracer or competitive ELISA using recombinant form of PD-1. Without this data, the loss of EH12.1 antibody binding in the context of Nivolumab presence cannot be interpreted.
22. I strongly disagree with multiple mentions that a part of important and relevant data is not presented – see below:
Row 310-312: “By contrast, there were no differences in the secretion of either IP-10 (CXCL10) and CCL5 (Rantes) (Figure 4B and 4D) or for IFNα, IL6 and TNFα secretion (data not shown).” - Why some data sets are presented and the other ones not?
Row 324-326: “We found that the PD1 expression on T cells was lost after Nivolumab treatment confirming its effect (Figure 5A). Moreover, the effect of Vesatolimod in the TLR-7 pathway was confirmed by the increased secretion of IFNα, IL6, TNFα and IL1β (Data not shown).”
Row 358-359: “At 48h after co-culture, the IL2 secretion was increased when TMEP-B or TMEP-Bmod were combined with Nivolumab, Vesatolimod or both together (data not shown).
I encourage the authors to provide the particular data sets into the supplementary file as I think those are important. If such data collections cannot be convincingly presented or are not substantial for the study, they should be removed from the manuscript.
33. In the all submitted manuscript, there are frequent typing errors, such as p values (0,05) see rows 250, 283, 296, 305, 308, 309 etc.
Reviewer 3 Report
This study provides evidence that, in vitro, transduction of induced dendritic cells with mRNA encoding multiple HIV-derived peptide epitopes recognized by CD4+ and CD8+ T cells in conjunction with either or both of two immunomodulators increases measures of T cell response. The two immunomodulators used are Vesatolimod, a TLR7 agonist, and Nivolumab, a T-cell checkpoint inhibitor. Some of the functions enhanced through this approach have previously been shown to correlate with viral control.
The authors should respond to the following comments.
1. In Figure 4, statistically significant differences in mean values for secretion of various mediators need to be interpreted cautiously given that for several analytes one to a few outlier points appear to dominate. The ranges of values for mock vs. mRNA groups generally overlap a great deal.
2. Can the authors offer any insights into why, in Figure 5, Vesatolimod seems as good as or better than the combination of Vesatolimod and Nivolumab for proliferation of T cells but not for secretion of IFNgamma, IP-10, or MIP-1beta?
3. Infection with HIV is associated with tissue damage or dysfunction resulting from excessive inflammation. Have the authors given thought to potential risks of autoimmunity or excessive inflammation after treating HIV-infected patients with Vesatolimod and/or Nivolumab?
4. In the Discussion, I believe it would be reasonable for the authors to acknowledge reports of serious immunopathological side effects of anti-checkpoint therapy in the context of treating cancer patients.
Minor issues:
Throughout the manuscript, there are issues related to word sequence (see example below), word choice, and use of plural nouns where singular nouns are typical, e.g., line 204: “DCs:T cells co-culture…,” which should be “DC:T cell co-culture.”
p.10, line 333 – “difference greater” would be better as “greater difference
“
